

# Morphological characterization and sexual dimorphism of the antennal sensilla in *Bactericera gobica* Loginova (Hemiptera: Psyllidae)—a scanning and transmission electron microscopic study

Yang Ge[1], Olivia M. Smith[2], Weilin Chen[3], Pingping Liu[4], Qingjun Yuan[1], Chuanzhi Kang[1], Tielin Wang[1], Jiahui Sun[1], Binbin Yan[1], Xiaoli Liu[5] and Lanping Guo[1]

[1] National Resource Center for Chinese Materia Medica, China Academy of Chinese Medical Sciences, State Key Laboratory Breeding Base of Dao-di Herbs, Beijing, China
[2] Ecology, Evolution, and Behavior Program, Michigan State University, East Lansing, MI, United States of America
[3] Pharmacy Faculty, Hubei University of Chinese Medicine, Wuhan, China
[4] Plant Protection Research Institute, Guangdong Provincial Key Laboratory of High Technology for Plant Protection, Guangzhou, China
[5] Plant Protection Research Institute, Ningxia Academy of Agricultural and Forestry Science, Ningxia, China

Corresponding authors
Xiaoli Liu, xiaoli_8302@163.com
Lanping Guo, glp01@126.com

## ABSTRACT

*Bactericera gobica* is the major pest of Goji berry plants and causes severe damage. Psyllids mainly use the antennal sensilla to recognize olfactory cues necessary to find host plants and mates. However, the structure and function of the antenna and the antennal sensilla of *B. gobica* remains previously unexplored. Here, we identify the external and internal morphology of the antennal sensilla of *B. gobica* using both scanning electron microscopy (SEM) and transmission electron microscopy (TEM). We found seven types of sensilla on the filiform antennae, including apical setae (LAS, SAS), sensilla basiconica (SB1, SB2), sensilla campaniform (SCA), sensilla chaetica (ChS1, ChS2), cavity sensilla (CvS1, CvS2), antennal rhinaria (AR1, AR2), and sensilla trichodea (ST). Five of these sensilla types—apical setae, sensilla basiconica, sensilla chaetica, cavity sensilla, and antennal rhinaria—may have olfactory functions based on their porous surfaces and internal dendritic outer segments (DOS). We also found several differences between the two sexes of *B. gobica* in the sensilla array and internal structure. ChS and DOS in the protrusions of AR were more abundant in males than females. Altogether, we comprehensively revealed the fine structure and probable function of *B. gobica* antennae and identified differences in the distribution and structure between psyllid sexes. Our findings provide important insights for future studies on defining the olfactory function of psyllid antenna using electrophysiological methods.

## INTRODUCTION

Goji berries are classified as superfruits because of the abundant bioactive compounds and nutrients in them (*Kulczynski & Gramza-Michalowska, 2016*). Goji berry psyllid *Bactericera gobica* (Hemiptera: Psyllidae) is one of the most widely distributed and devastating pests of goji berry plants, causing severe direct and indirect damage (*Wu et al., 2017*). *B. gobica* can induce damage directly by sucking the leaves and phloem sap of goji berry plants, resulting in the premature defoliation of the leaves, and sometimes the death of the whole plant (*Li et al., 2018*). Additionally, the accumulation of the sugar-rich honeydew secreted by *B. gobica* can cause indirect damage by facilitating the growth of sooty mold on leaves. *B. gobica* has also been demonstrated to vector another important pest of goji, the gall mite *Aceria pallida* (*Liu et al., 2019*). Thus, control of *B. gobica* is of great importance to the production of goji berries. Control of *B. gobica* using pesticides and natural enemies has had some success (*Wu et al., 2017*). However, chemical control is not sustainable due to pesticide resistance, and the use of natural enemies is not sufficient due to *B. gobica*'s high fecundity.

Olfactory cues play an important role in host recognition, mating, and oviposition by psyllids (*Kristoffersen, Larsson & Anderbrant, 2008*). Antennae are peripheral sensory structures and the main olfactory sensory organs of psyllids. Psyllid antenna include various types of sensory sensilla that have mechanosensory, thermo-hygroreceptive, and chemosensory functions, such as detection of various stimuli involved in host and mate location. Olfactory-based pest control strategies have been studied in several psyllid species. For instance, the pear psyllid *Cacopsylla pyri* and *C. pyricola* (*Ganassi et al., 2018*; *Guédot, Horton & Landolt, 2009*) and Asian citrus psyllid *Diaphorina citri* (*Zanardi et al., 2018*) both use olfactory cues to locate host plants and mates. Accordingly, the development of behavioral manipulation control methods that disrupt olfactory cues could be promising.

Despite the economic importance of *B. gobica*, the potential for semiochemical-based monitoring or control methods has not yet been investigated because crucial background information has been missing. The use of scanning and transmission electron microscopy enables researchers to infer the probable olfactory reception of organisms based on morphology (*Zhang et al., 2020*). Nymph and adult antennae of several psyllid species, including Asian citrus psyllid *D. citri* (*Zheng et al., 2020*), potato psyllid *Bactericera cockerelli* (*Arras, Hunter & Bextine, 2012*), and carrot psyllid *Trioza apicalis* (*Kristoffersen et al., 2006*), have been investigated with scanning electron microscopy (SEM). These studies have revealed several non-porous and porous sensilla on the psyllid antennae. However, very few studies have conducted more detailed morphological investigations to reveal the inner features of antennal sensilla that can best be shown by transmission electron microscopy (TEM) (*Kristoffersen et al., 2006*). The classification of antennal sensilla can be very difficult using external morphology alone, and external structures are not reliable enough to interpret the function of different sensilla (*Kristoffersen et al., 2006*; *Onagbola et al., 2008*). To date, no studies have been published documenting the morphology of *B. gobica* antennae and antennal sensilla.

Here, we begin the process of defining the olfactory capacities of the goji berry psyllid *B. gobica* by describing its antennal sensilla. We identify and describe the array and morphology of the sensilla of goji berry psyllid antennae. We then suggest possible olfactory functions of sensillae using the external (*via* SEM) and internal (*via* TEM) morphological features we documented in goji berry psyllid.

## MATERIALS & METHODS

### Insects

The goji berry psyllid *B*. *gobica* colony used for this study was originally from field-collected individuals harvested from goji berry orchards in Zhongwei City (7°17′42″N, 105°38′7″E) and Yinchuan City (38°38′23″N, 106°7′13″E), Ningxia Province, China. The psyllid colony was maintained on potted goji berry plants (*Lycium barbarum* L.) in growth chambers under controlled conditions (25 ± 2 °C, 70 ± 5% humidity, 16L:8D photoperiod). Goji berry plants were replaced every two days to allow for oviposition, and plants with *B. gobica* eggs were transferred to nymph cages until hatching. Newly emerged *B. gobica* adults were collected twice a day from nymph cages and transferred to individual cages for development until 3-d old (*i.e.,* when they generally reach reproductive maturity and begin ovipositing). At 3-d old, adults that had reached reproductive maturity were prepared for scanning electron microscopy and transmission electron microscopy analyses.

### Scanning electron microscopy (SEM)

Sixteen specimens of each sex (16 male, 16 female) of 3-day old adult *B. gobica* were analyzed using SEM. We cleaned the entire bodies of the *B. gobica* specimens two times each at 70 W for 5s in 70% ethanol using ultrasonic waves. Next, the specimens were dehydrated in a graded series of 80%, 90%, and 99.9% ethanol for 20 min at each concentration before being transferred to a mixed solution of ethanol and tert-butyl alcohol (1:1, by volume) for 20 min. Specimens were then postfixed in tert-butyl alcohol and vacuum freeze-dried following the methods used in *Hao, Sun & Liu (2020)*. The antennae were mounted in ventral, dorsal, posterior, and anterior positions on SEM stubs (*Fu et al., 2012*) with double-sided copper sticky tape, and sputter-coated using a high-resolution sputter coater (Hitachi, Japan). Finally, the specimens were examined using an S-3400 scanning electron microscope (Hitachi, Japan) at 15.0 kV.

### Transmission electron microscopy (TEM)

We prepared five samples of each sex (five male, five female) of *B. gobica* for TEM following the methods described in *Kristoffersen et al. (2006)* and *Onagbola et al. (2008)*. We fixed the *B. gobica* overnight in 2.5% glutaraldehyde and 100% Tween 20 in 0.1M cacodylate buffer. Specimens were rinsed in the buffer and post-fixed in 1% osmium tetroxide for 2 h, dehydrated in a graded series of ethanol. Specimens were sequentially infiltrated in acetone, resin (2:1, 1:1), and pure resin at 32 °C then embedded in pure 812# resin. Specimens were then polymerized sequentially at 37 °C, 50 °C, and 60 °C for 12 h at each temperature. Ultrathin 80–100 section were cut with a glass knife, mounted on copper grids, and then stained with uranyl acetate and lead citrate for 15–20 min. The sections were observed in a JEOL transmission electron microscope (JEOL Ltd., Japan) operated at 80 kV.

## Structure analysis

There have been inconsistencies in the terminology and descriptions of insect antennal sensilla. Here, we classified and named the antennal sensilla of *B. gobica* following the nomenclature of *Schneider (1964)*, *Zacharuk (1980)*, *Kristoffersen et al. (2006)*, *Onagbola et al. (2008)*, and *Zheng et al. (2020)*, alongside the morphological details we observed through our SEM and TEM analysis.

## Data analysis

We used the software ImageJ (https://imagej.nih.gov/ij/) to measure the length of each segment of *B. gobica* antennae; the size of different types of sensilla; and the length, basal width, and apical width of sensilla chaetica (ChS) on different antennal segments. We used one-way ANOVAs with Tukey post hoc comparisons to compare the length of the segments and the size of different types of sensilla. Two sample t-tests were used to compare the size of each individual sensilla subtype between the two sexes in the software SPSS 26.0. We used a Bonferroni correction to determine our alpha value ($\alpha$) for assessing statistical significance for the size of different types of sensilla and sensilla chaetica on different segments. We used a Bonferroni correction to avoid the inflation of type I errors from making multiple comparisons. Our $\alpha$ was set at $P = 0.017$ ($\alpha = 0.05/3$ comparisons [the length, basal width, apical width]). We present mean lengths $\pm$ standard error below.

# RESULTS

## General description of the antennae

The external shape of the female and male adult antennae were nearly identical. The filiform antennae of both sexes was located in front of their compound eyes, consisting of a basal scape, a pedicel, and eight flagellomeres (F1-F8) (Fig. 1). All segments were cylindrical, and the width of each flagellomere was relatively uniform from bottom to top. The total length of the male antenna ($1004.782 \pm 20.007\,\mu$m) was slightly longer than the female antenna ($940.306 \pm 37.136\,\mu$m), though the difference was not statistically significant (Two-sample $t$-test, $t_{28} = -1.528$, $p = 0.144$). The scape, pedicel, and flagellum accounted for an average of 5.184%, 4.592%, and 90.224% of the total length of female antennae, respectively, and 4.151%, 4.082%, and 91.767% of the total length of male antennae, respectively. The first flagellomere (F1) was the longest segment in both sexes of *B. gobica* (male: $213.339 \pm 9.135$ $\mu$m and female: $199.920 \pm 13.411$ $\mu$m). We found no significant differences between female and male antenna lengths in any of the 10 individual antenna segments, based on two-sample t-tests (Table 1).

## Antennal sensilla

The surface of *B. gobica* antennae was scaly with several types of sensilla distributed across it. SEM and TEM imaging revealed seven types of antennal sensilla in total, including sensilla trichodea (ST), apical setae (AS), sensilla basiconica (SB), sensilla chaetica (ChS), sensilla campaniform (SCA), cavity sensilla (CvS), and antennal rhinaria (AR). Among the seven types of antennal sensilla, AS, ChS, SB, AR, and CvS could each be further divided into two subtypes (Table 2). Two hair-shaped sensilla, ST and ChS, were found on the scape

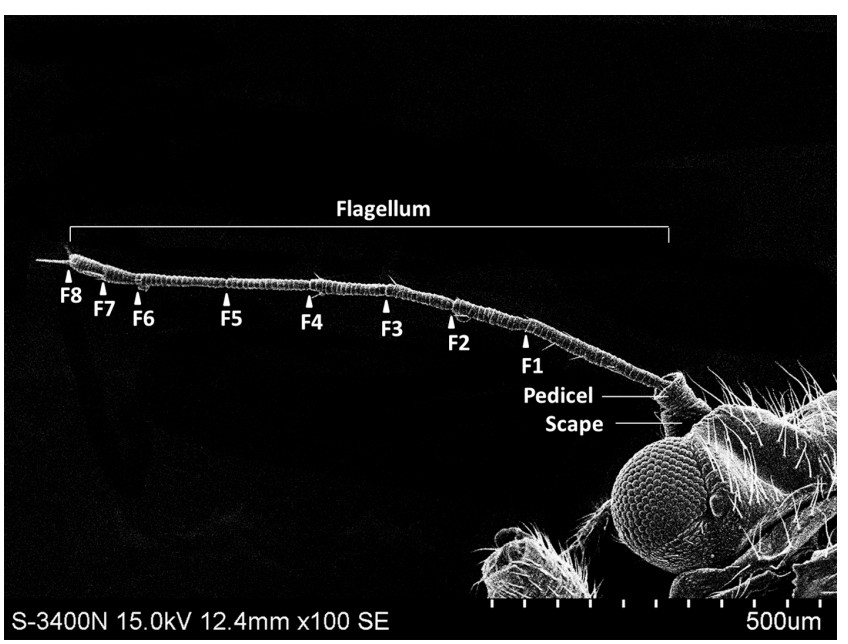

**Figure 1** **The whole view of antennal segments of *Bactericera gobica*.** It shows the posterior dorsal field of the left antenna of *B. gobica*. F1–F8, antennal flagellomeres 1-8.

**Table 1** **The length of each antennal segment of female and male *Bactericera gobica*.**

| Sensilla type | Length (μm) | | Sample size | *df* | *T*-value | *P*-value |
|---|---|---|---|---|---|---|
| | Female | Male | | | | |
| Scape | 48.741 ± 3.877 | 41.706 ± 6.810 | 17 | 15 | 0.924 | 0.370 |
| Pedicel | 43.182 ± 2.123 | 41.020 ± 2.344 | 23 | 21 | 0.685 | 0.501 |
| F1 | 199.920 ± 13.411 | 213.339 ± 9.135 | 23 | 21 | −0.812 | 0.426 |
| F2 | 108.905 ± 3.438 | 113.933 ± 4.290 | 30 | 28 | −0.915 | 0.369 |
| F3 | 111.179 ± 2.702 | 106.777 ± 4.626 | 28 | 26 | 0.869 | 0.393 |
| F4 | 125.918 ± 2.915 | 138.044 ± 8.148 | 25 | 23 | −1.534 | 0.139 |
| F5 | 128.895 ± 3.632 | 134.589 ± 4.780 | 26 | 24 | −0.963 | 0.345 |
| F6 | 119.452 ± 5.624 | 136.136 ± 6.948 | 24 | 22 | −1.881 | 0.073 |
| F7 | 50.261 ± 2.100 | 56.214 ± 7.265 | 26 | 24 | −0.895 | 0.379 |
| F8 | 49.055 ± 1.60 | 45.185 ± 2.175 | 30 | 28 | 1.464 | 0.158 |
| Total | 940.306 ± 37.136 | 1004.782 ± 20.007 | 20 | 18 | −1.528 | 0.144 |

**Notes.**

Mean ± standard error (SE); *T*-value and *P*-value shown for pairwise comparisons (from two-sample *t*-tests) between the length of female and male antennal segment.

segment, and one more type of sensilla, SCA, was found on the pedicel segment (Table 2). The sensilla types on flagellomeres were more diverse than those on the scape and pedicel and included AS, SB, ChS, SCA, CvS, and AR. The quantity of sensilla was similar from F2 through F6 then increased from F7 to F8. Antennal sensilla were intensively arranged on F1 and the terminal segment F8. The outer view of the scape had a greater abundance and
**Table 2 Number and distribution of antennal sensilla in female and male *Bactericera gobica*.** If the number varied between individual *B. gobica* examined, numbers are given as a range.

| Segment | | Sex | Sensilla Trichodea ST | Apical setae | | Sensilla Basiconica | | Sensilla Chaetica | | Sensilla Campaniform | Cavity Sensilla | | Antennal Rhinaria | | Sample size |
|---|---|---|---|---|---|---|---|---|---|---|---|---|---|---|---|
| | | | | LAS | SAS | SB-1 | SB-2 | ChS-1 | ChS-2 | SCA | CvS-1 | CvS-2 | AR-1 | AR-2 | (Psyllid No.) |
| Scape | | Male | 3 | – | – | – | – | 3 | – | – | – | – | – | – | – |
| | | Female | 3 | – | – | – | – | 3 | – | – | – | – | – | – | – |
| Pedicel | | Male | 2 | – | – | – | – | – | 3 | 1 | – | – | – | – | – |
| | | Female | 2 | – | – | – | – | – | 4 | 1 | – | – | – | – | – |
| Flagellum | F1 | Male | – | – | – | – | – | – | 7 | – | – | – | – | – | – |
| | | Female | – | – | – | – | – | – | 6 | – | – | – | – | – | – |
| | F2 | Male | – | – | – | – | – | – | 2 | – | – | – | 1 | – | – |
| | | Female | – | – | – | – | – | – | 2 | – | – | – | 1 | – | – |
| | F3 | Male | – | – | – | – | – | – | 1–2 | – | – | – | – | – | – |
| | | Female | – | – | – | – | – | – | 1 | – | – | – | – | – | – |
| | F4 | Male | – | – | – | – | – | – | 2 | – | – | – | 0–1 | 0–1 | – |
| | | Female | – | – | – | – | – | – | 2–3 | – | – | – | 0–1 | 0–1 | – |
| | F5 | Male | – | – | – | – | – | – | 1 | – | – | – | – | – | – |
| | | Female | – | – | – | – | – | – | 1 | – | – | – | – | – | – |
| | F6 | Male | – | – | – | – | – | – | 2–3 | – | – | – | 0–1 | 0–1 | – |
| | | Female | – | – | – | – | – | – | 2 | – | – | – | 0–1 | 0–1 | – |
| | F7 | Male | – | – | – | 1 | 1 | – | 1–3 | – | – | – | – | 1 | – |
| | | Female | – | – | – | 1 | 1 | – | 1 | – | – | – | – | 1 | – |
| | F8 | Male | – | 1 | 1 | – | – | – | 1 | – | 1 | 1 | – | – | – |
| | | Female | – | 1 | 1 | – | – | – | 1 | – | 1 | 1 | – | – | – |
| Total | | Male | 5 | 1 | 1 | 1 | 1 | 3 | 20–24 | 1 | 1 | 1 | 4 | | 15 |
| | | Female | 5 | 1 | 1 | 1 | 1 | 3 | 20–21 | 1 | 1 | 1 | 4 | | 16 |

Notes.

ST, sensilla trichodea; ChS-1 and ChS-2 are sensilla chaetica type 1 and 2, respectively; SB-1 and SB-2 are sensilla basiconica type 1 and 2, respectively; LAS and SAS are long and short apical setae, respectively; AR-1 and AR-2 are antennal rhinaria type 1 and 2, respectively; CvS-1 and CvS-2 are cavity sensilla type 1 and 2, respectively; SCA, sensilla campaniform; '-' indicates absent.

diversity of sensilla than the inner view. We found this in both sexes. The distributions and morphological characteristics of the antennal sensilla are summarized in Tables 2 and 3.

## Sensilla trichodea (ST)

In both sexes, sensilla trichodea (ST) were mostly distributed on the outer lateral side of the dorsal view of the scape segment (Figs. 2A and 2G) and the inner margin of the anterior dorsal view of the pedicel segment (Fig. 2B). ST were slender, long, and strongly curved, with a slightly grooved surface, pointed tip, and in a slightly concave shallow socket (Figs. 2G and 2H). The length was about 21.871 µm for females and 22.307 µm for males, and the basal width was about 1.487 µm for females and 1.503 µm for males (Table 4). TEM analysis showed that the ST were single walled (SW) sensilla. We found no pores penetrating the cuticular walls, indicating that ST are aporous sensilla (Fig. 3A).

## Sensilla chaetica (ChS)

Sensilla chaetica (ChS) were the most widely distributed and abundant sensilla type on the antenna of *B. gobica*. We found ChS on every segment in both males and females.

**Table 3  Main features and probable function of *Bactericera gobica* antennal sensilla.**

| | Description | | | | | Number of DOS | | Function | |
|---|---|---|---|---|---|---|---|---|---|
| | Porosity | Tip | Wall | Shape | Socket | Female | Male | Female | Male |
| ST | Aporous | Sharp | Grooved | Strongly curved | unobvious | 1 | 1 | Non-olfactory | Non-olfactory |
| ChS-1 | Multiporous | Sharp | Grooved | Strongly curved | obvious | 1 | 1 | Olfactory | Olfactory |
| ChS-2 | Multiporous | Sharp | Grooved | Straight or slightly curved | obvious | 1 | 1 | Olfactory | Olfactory |
| SB-1 | Multiporous | Blunt | Grooved | Straight | unobvious | 4 | 3 | Olfactory | Olfactory |
| SB-2 | Multiporous | Blunt | Grooved | Straight | unobvious | 3 | 3–7 | Olfactory | Olfactory |
| LAS | Multiporous | Blunt | Grooved | Straight | obvious | 45 | 18 | Olfactory | Olfactory |
| SAS | Multiporous | Blunt | Grooved | Straight | obvious | 21 | <18 | Olfactory | Olfactory |
| AR-1 | Multiporous | – | Pitted | protrusion | – | multi | multi | Olfactory | Olfactory |
| AR-2 | Multiporous | – | – | Cavity shape | – | multi | multi | Olfactory | Olfactory |
| CvS-1 | Multiporous(female) aporous(male) | – | – | – | – | – | – | Olfactory | Non-olfactory |
| CvS-2 | Aporous | – | – | – | – | – | – | Non-olfactory | Non-olfactory |
| SCA | Aporous | – | Smooth | – | – | – | – | Non-olfactory | Non-olfactory |

ChS constituted about 46.51–61.54% of the total number of antennal sensilla (Table 2). We found ChS alone, in pairs, or in groups of three on the distal part of each antennal flagellomere from F2 through F7 (Figs. 4A–4L). ChS occurred on the medial portions of the scape, pedicel, and F1 (Figs. 2B–2D). One ChS was identified below the base of the long apical setae on F8 (Fig. 5A). Based on the morphology, length, and location of the sensilla, we divided ChS into two subtypes. The length and width of the two ChS subtypes on each antennal segment of female and male *B. gobica* are presented in Table 5.

ChS-1 had a very similar conformation to the ST and were strongly curved with tight and obvious sockets (Fig. 2C). ChS-1 were distributed on the outer lateral side of the dorsal view of the scape segment and had a sharp tip and grooved surfaces. The length of ChS-1 was about 27.339 μm for females and 28.756 μm for males, and the basal width was about 1.760 μm for females and 1.515 μm for males.

ChS-2 were straight, with sharp or slightly curved tips, grooved surfaces, and situated in tight and obvious sockets (Figs. 2D–2F). ChS-2 were distributed on the anterior dorsal view of the pedicel segment, on the anterior lateral part of F2–F8, and were evenly distributed on F1. ChS-2 were shorter than ChS-1. The average length of ChS-2 was about 19.247 μm for females and 19.681 μm for males, and the basal width was about 1.891 μm for females and 2.131 μm for males. In both sexes, the length of ChS-2 on F1–F3 was significantly longer than that on F6–F8 (Female: one-way ANOVA, $F_9 = 11.967$, $P = 0.000$; Male: one-way ANOVA, $F_9 = 10.018$, $P = 0.000$) (Table 5). ChS-2 on F7 and F8 was the shortest (Table 5). The SEM analysis revealed two obvious pores on one of the ChS-2 (Fig. 2F), while no other differences in the inner structure were found between the ChS-2 sensilla with pores and those without.

Despite morphological differences between the two subtypes of ChS described above, TEM analysis showed both subtypes were single walled (SW). The cross sections of ChS-1

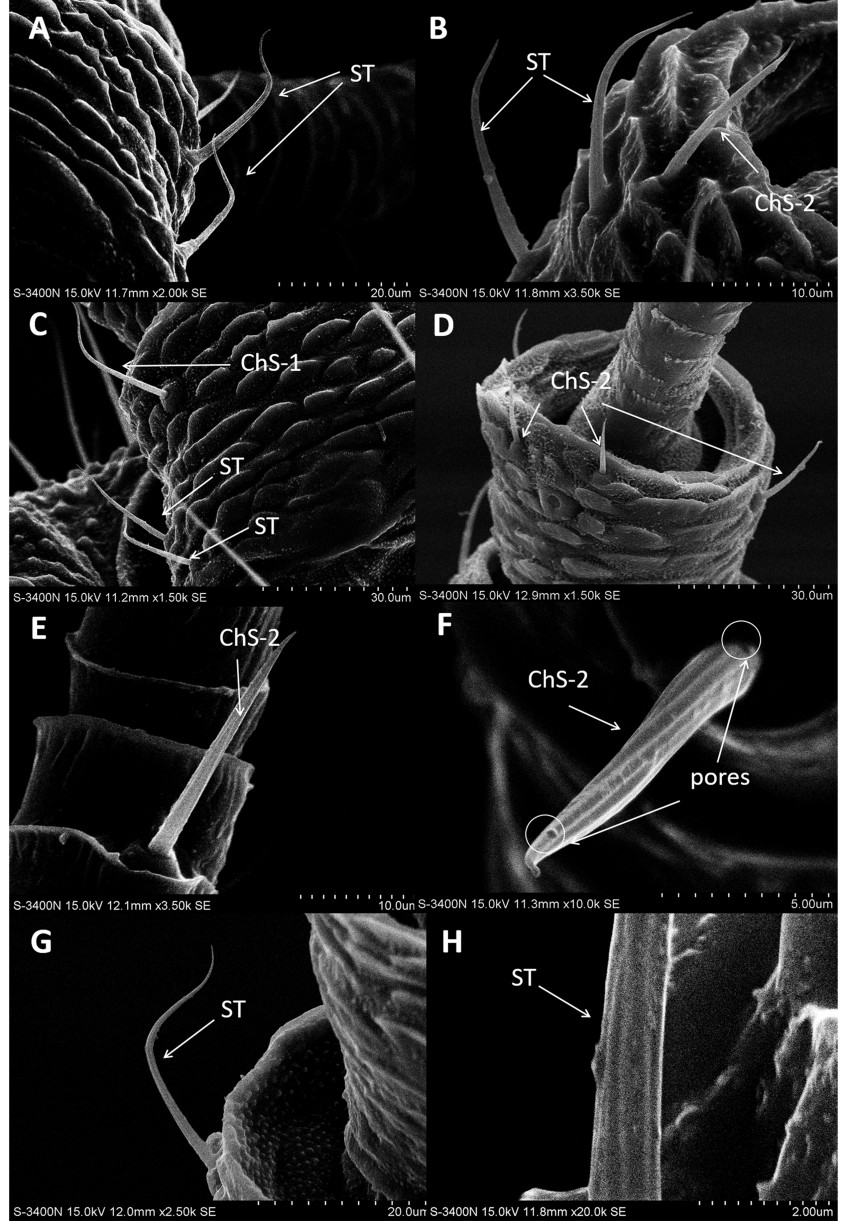

**Figure 2** Scanning electron micrographs of various antennal sensilla of *Bactericera gobica*. (A) Strongly curved sensilla trichodea (ST) on scape with sharp tip. (B) ST and sensilla chaetica subtype 2 (ChS-2) with obvious socket on pedicel. (C) Strongly curved sensilla chaetica subtype 1 (ChS-1) with obvious socket and ST with unobvious socket on scape. (D) Straight or slightly curved ChS-2 on pedicel. (E) Straight ChS-2 with grooved surface and sharp tip on flagellomere 3. (F) Porous ChS-2 with grooved surface. (G) Strongly curved ST on scape. (H) High magnification of the grooved surface of ST.

and ChS-2 were either septilateral or octagon shaped (Figs. 3B–3C). However, the different shapes of the cross sections of ChS might simply be due to the different levels of the sections. Pores were found penetrating the cuticular walls in all the ChS indicating that ChS were porous sensilla and may have olfactory function (Figs. 3B–3C). The more proximal cross

**Table 4  The size of antennal sensilla in female and male *Bactericera gobica*.**

| Sensilla type | Length (μm) | | Sample size | T value | P value | Basal width/Width (μm) | | Sample size | T value | P value | Apical width (μm) | | Sample size | T value | P value |
|---|---|---|---|---|---|---|---|---|---|---|---|---|---|---|---|
| | Female | Male | | | | Female | Male | | | | Female | Male | | | |
| ST | $21.871 \pm 1.400$ | $22.307 \pm 1.213$ | 29 | $T_{27}=-0.223$ | 0.826 | $1.487 \pm 0.135$ | $1.503 \pm 0.037$ | 29 | $T_{27}=-0.114$ | 0.910 | $0.407 \pm 0.033$ | $0.410 \pm 0.027$ | 29 | $T_{27}=-0.061$ | 0.952 |
| ChS-1 | $27.339 \pm 2.978$ | $28.756 \pm 1.510$ | 8 | $T_{6}=-0.344$ | 0.743 | $1.760 \pm 0.081$ | $1.515 \pm 0.158$ | 8 | $T_{6}=1.548$ | 0.173 | $0.444 \pm 0.088$ | $0.380 \pm 0.048$ | 8 | $T_{6}=0.528$ | 0.616 |
| ChS-2 | $19.247 \pm 0.567$ | $19.681 \pm 0.509$ | 316 | $T_{314}=-0.570$ | 0.569 | $1.891 \pm 0.049$ | $2.131 \pm 0.190$ | 191 | $T_{189}=-1.174$ | 0.242 | $0.519 \pm 0.025$ | $0.565 \pm 0.048$ | 191 | $T_{189}=-0.827$ | 0.410 |
| SB-1 | $14.434 \pm 2.454$ | $13.577 \pm 3.345$ | 14 | $T_{12}=0.207$ | 0.841 | $2.443 \pm 0.111$ | $2.577 \pm 0.208$ | 10 | $T_{8}=-0.568$ | 0.586 | $0.914 \pm 0.058$ | $0.863 \pm 0.054$ | 10 | $T_{9}=0.642$ | 0.538 |
| SB-2 | $29.561 \pm 1.816$ | $24.425 \pm 2.163$ | 17 | $T_{15}=1.793$ | 0.093 | $3.102 \pm 0.235$ | $3.391 \pm 0.334$ | 16 | $T_{14}=-0.666$ | 0.516 | $0.969 \pm 0.084$ | $0.956 \pm 0.079$ | 16 | $T_{14}=0.105$ | 0.918 |
| LAS | $51.306 \pm 0.899$ | $48.840 \pm 1.629$ | 22 | $T_{20}=1.429$ | 0.169 | $5.168 \pm 0.206$ | $5.813 \pm 0.089^{*}$ | 22 | $T_{20}=-2.881$ | 0.012 | $2.211 \pm 0.221$ | $2.233 \pm 0.067$ | 22 | $T_{20}=-0.088$ | 0.931 |
| SAS | $18.575 \pm 1.825$ | $16.960 \pm 1.954$ | 21 | $T_{19}=0.597$ | 0.557 | $5.023 \pm 0.351$ | $5.537 \pm 0.110$ | 20 | $T_{18}=-1.399$ | 0.179 | $2.713 \pm 0.232$ | $2.631 \pm 0.091$ | 20 | $T_{18}=0.276$ | 0.786 |
| AR-1 | $19.250 \pm 1.488$ | $15.929 \pm 1.481$ | 50 | $T_{48}=1.581$ | 0.120 | $13.896 \pm 1.623$ | $11.331 \pm 0.826$ | 42 | $T_{40}=1.408$ | 0.170 | – | – | | – | – |
| AR-2 | $7.049 \pm 2.666$ | $10.368 \pm 2.042$ | 13 | $T_{11}=-1.004$ | 0.337 | $5.176 \pm 2.443$ | $5.213 \pm 1.615$ | 14 | $T_{12}=-0.130$ | 0.990 | – | – | | – | – |
| CvS-1 | $1.757 \pm 0.197$ | $1.519 \pm 0.195$ | 9 | $T_{7}=0.848$ | 0.424 | $0.923 \pm 0.155$ | $1.140 \pm 0.251$ | 9 | $T_{7}=-0.687$ | 0.514 | – | – | | – | – |
| CvS-2 | $2.198 \pm 0.272$ | $1.546 \pm 0.261$ | 10 | $T_{8}=1.729$ | 0.122 | $1.201 \pm 0.084$ | $1.524 \pm 0.201$ | 10 | $T_{8}=-1.481$ | 0.177 | – | – | | – | – |
| SCA | $8.154 \pm 0.350$ | $7.643 \pm 0.771$ | 11 | $T_{9}=0.562$ | 0.588 | $6.558 \pm 0.421$ | $6.411 \pm 0.455$ | 11 | $T_{9}=0.232$ | 0.821 | – | – | | – | – |

**Notes.**

Asterisk * indicates a significant difference between female and male psyllids (we used two sample t-tests and assessed significance using a Bonferroni correction (alpha = 0.053 comparisons = 0.017). $T$-value and $P$-value for pairwise comparisons between female and male sensilla types shown next to mean $\pm$ SE.

section of the basal part of ChS showed this sensilla was innervated by one sensory neuron (Figs. 3D–3E). Individual differences in the number of ChS-2 on F3, F4, F6 and F7 was found in both female and male *B. gobica* (Table 2).

## Sensilla apical setae (LAS and SAS)

A long apical setae (LAS) and a short apical setae (SAS) were situated apically on the *B. gobica* antenna. The LAS was distributed on the outside of F8 of the dorsal view compared to the SAS (Figs. 5A and 5D). Longitudinally arranged grooves were found on both LAS and SAS. The shafts of LAS were straight or slightly curved, while SAS were straight. The tips of LAS and SAS were both blunt (Figs. 5A and 5D). The LAS was about $51.306 \pm 0.899$ μm long for females and $48.840 \pm 1.629$ μm long for males. The SAS was about $18.575 \pm 1.825$ μm long for females and $16.960 \pm 1.954$ μm long for males.

TEM analysis showed that LAS and SAS were both SW and that there was a well-developed ring of sensillar channels in LAS and SAS, suggesting a plausible role in olfactory perception. No obvious dendritic outer segments (DOS) were found in the cross-section images at the tips of LAS and SAS (Fig. 6A). However, more DOS were found in cross-section images in the lower parts of LAS and SAS (Fig. 6B). Up to 45 DOS were found in LAS lumens of female *B. gobica* (Fig. S1), and up to 21 DOS were found in SAS lumens of female *B. gobica* (Fig. 6A2). Eighteen was the largest number of DOS that we found in male LAS cross sections (Fig. S2). We found more DOS in LAS than SAS. Correspondingly, two groups of dendrites with sheaths were found next to the cross sections of CvS-1, presumably innervating the LAS. Another group of dendrites with sheaths was found below the SAS where it connected to the tip of F8 (Figs. 6C–6D).

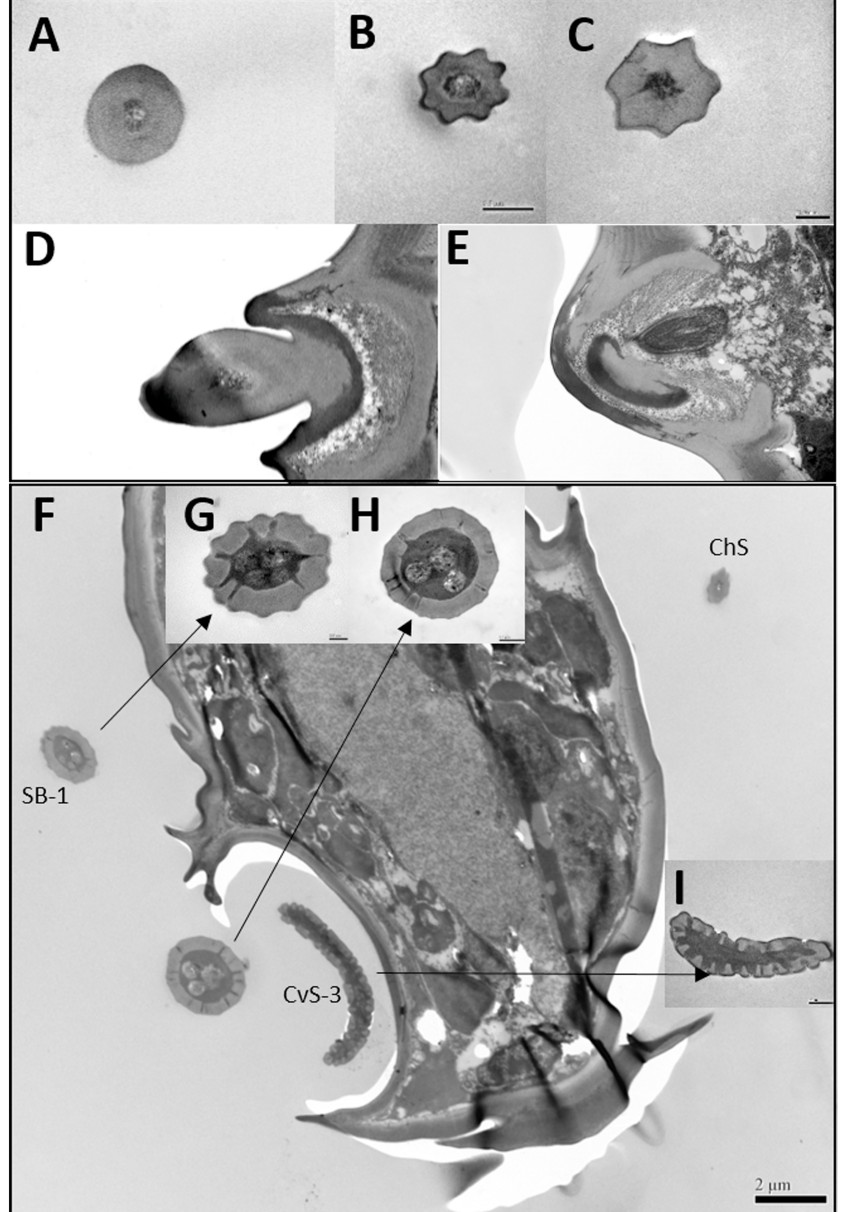

**Figure 3  Transmission electron micrographs of various antennal sensilla of *Bactericera gobica*.** (A) Cross section of ST, and there are no pores on ST. (B) Octagon shaped cross section of ChS, multiple pores penetrating the cuticle can be clearly seen in the cross section. (C) Septilateral transect of ChS with clear cuticle pores. (D–E) Cross section and a more proximal cross section of the basal part of ChS . (F) Overview of transect through AR-2 and adjoining sensilla basiconica SB1 and SB2 and sensilla chaetica ChS-2 on flagellomere 7, many dendritic outer segments (DOS) were found in the cross section of the protrusion in AR-2. (G) High magnification of the cross section of SB-1 with four DOS in the lumen and a well-developed ring of sensillar channels. (H) High magnification of the cross section of SB-2 with three DOS in the lumen and a well-developed ring of sensillar channels. SB-2 is positioned inside the opening of AR-2. (I) High magnification of the cross section of a more distal section of AR-2 showing no DOS at the upper part of the protrusion in AR-2.

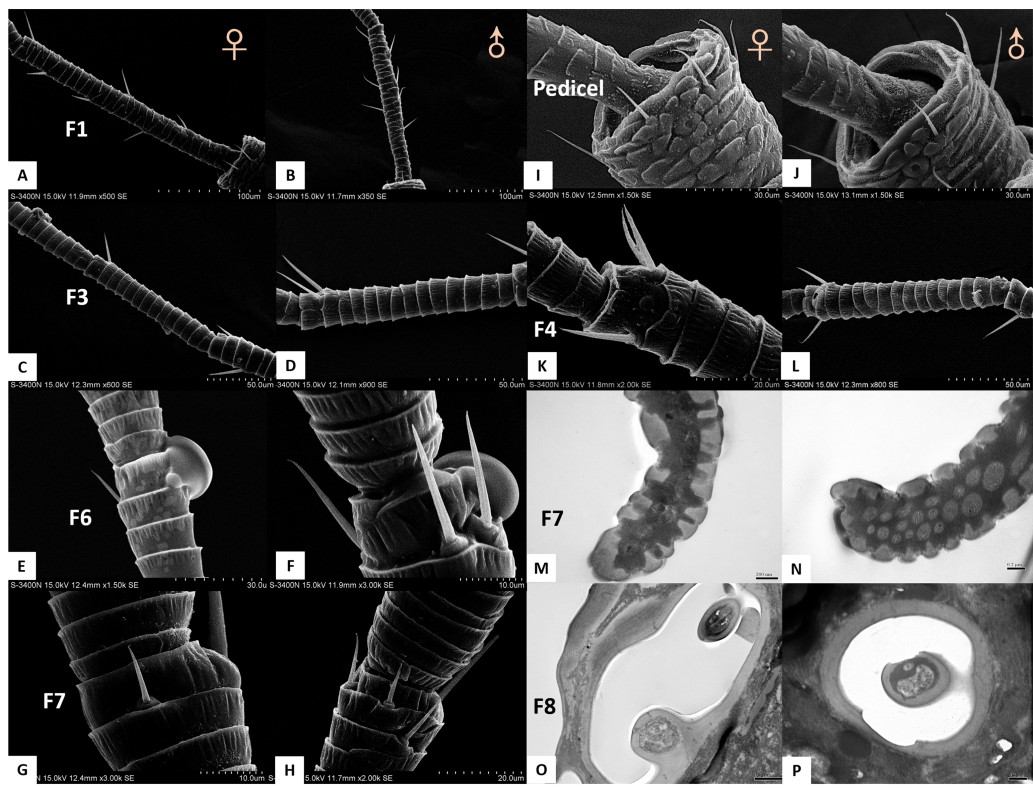

**Figure 4** **Differences between female and male antennal sensilla.** (A–H) Sensilla chaetica ChS set-up on flagellomeres F1 (A & B), F3 (C & D), F6 (E & F), and F7 (G & H) of females and males. Male antenna had one to two more flagellomeres than female antenna. (I–L) ChS set-up on pedicel segment (I & J) and F4 (K & L). Female antenna had one more ChS on the pedicel and F4 than male antenna. (M–N) Cross section of the protrusion in antennal rhinaria (AR) on F7. Fewer dendritic outer segments (DOS) were found in (M) females' antenna than (N) males' antenna. (O–P) Cross section of cavity sensilla type 1 (CvS-1). There is one more porous peg at the bottom of CvS-1 in female than male antenna.

## Sensilla basiconica (SB-1 and SB-2)

In both sexes, sensilla basiconica (SB) can only be found on the apical part of the ventral view of F7 (Figs. 5G–5H). The SB sensilla can be further divided into two subtypes (SB-1 and SB-2) based on their length and location (Figs. 4G and 4H). The two SB were cone-shaped, with blunt tips, and grooved surfaces. The SB were thicker than ST and ChS (Table 4).

There was only one SB-1 on the terminal part of F7. SB-1 averaged 14.434 ± 2.454 μm long for females and 13.577 ± 3.345 μm long for males. It was shorter than SB-2, which averaged 19.247 ± 0.567 μm for females and 19.247 ± 0.567 μm for males. SB-1 was located next to the antennal rhinaria type 2 (AR-2) and did not have any specialized basal membrane (Figs. 5G–5H). SB-2 protruded directly from the pit of AR-2. Our TEM analysis showed that SB had similar inner structures to apical setae, and we found well-developed pore tubules in the cross sections of SB (Figs. 3G–3H). Similar to apical setae, more DOS were found in the lower parts of SB. Four DOS were found in SB-1 (Figs. 3F–3G), and up to seven DOS were found in SB-2 (Fig. S3).

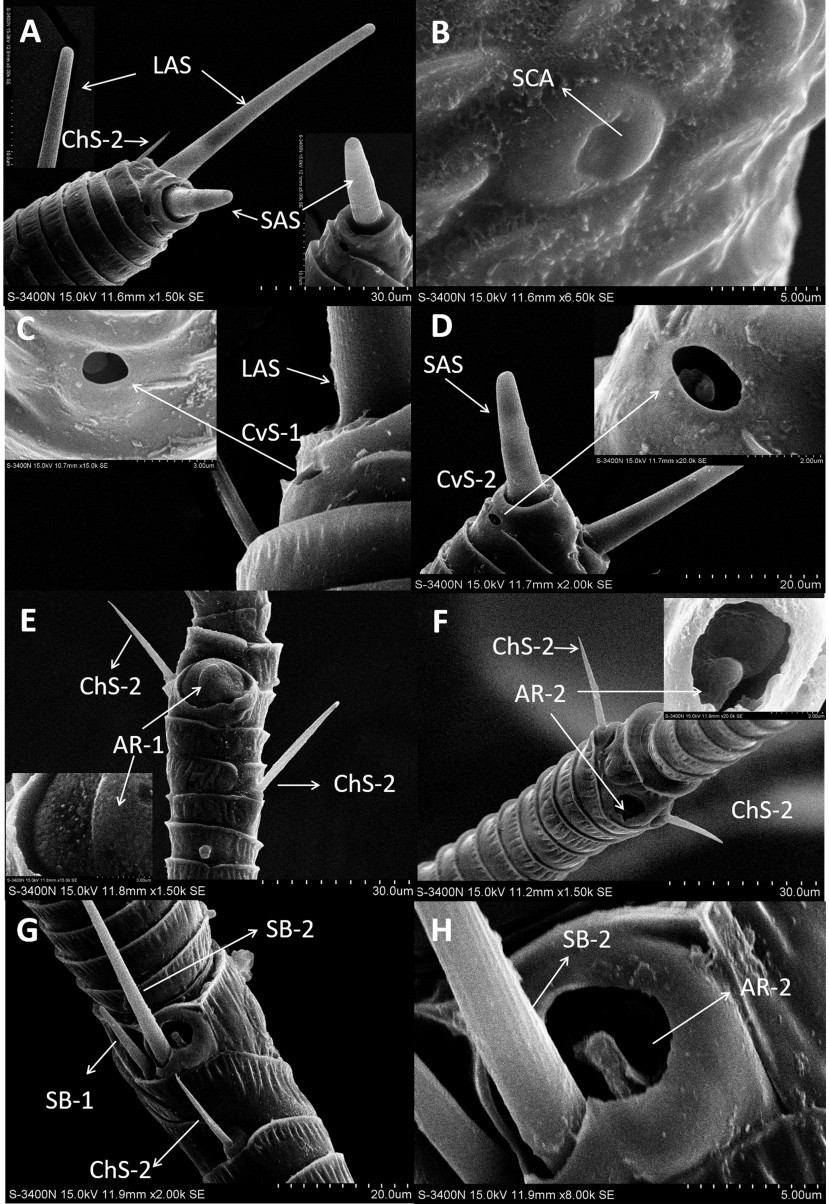

**Figure 5** **Scanning electron micrographs of various antennal sensilla of *Bactericera gobica*.** (A) Long apical setae (LAS) and short apical setae (SAS) with blunt tip and the magnification of LAS and SAS. (B) Aporous sensilla campaniform (SCA) on pedicel. (C) LAS and cavity sensilla subtype 1 (CvS-1) at the base of LAS on female *B. gobica* antenna. The high magnification of the opening of CvS-1 showed only one sensory peg in CvS-1, although two pegs were found at the bottom of CvS-1 by TEM analysis. (D) SAS and the cavity sensilla subtype 2 (CvS-2) at the base of SAS. The high magnification of the opening of CvS-2 showed similar external structure to CvS-1, but there is only one peg at the bottom of CvS-2. (E) Sensilla chaetica subtype 2 (ChS-2) and antennal rhinaria subtype 1 (AR-1) with pitted surface on flagellomere 2. (F) ChS-2 and a cavity shaped rhinaria subtype 2 (AR-2) on flagellomere 4. The high magnification of the opening of AR-2 showed a sensory peg inside. (continued on next page...)

**Figure 5 (…continued)**
(G) Two SB-1 and SB-2 with blunt tip and a ChS-2 sensilla with sharp tip on flagellomere 7. (H) High magnification of the grooved surface of SB-2 and AR-2 on flagellomere 7. SB-2 protrudes from the cavity of AR-2, and a large sensory peg can be seen at the opening of AR-2.

**Table 5 The size of sensilla chaetica in female and male *Bactericera gobica*.**

| Antennal segment | Sensilla chaetica (ChS) | Length (μm) | | Sample size | Basal width (μm) | | Sample size | Apical width (μm) | | Sample size |
|---|---|---|---|---|---|---|---|---|---|---|
| | | Female | Male | | Female | Male | | Female | Male | |
| Scape | ChS-1 | 27.339 ± 2.978a | 28.756 ± 1.510a | 8 | 1.76 ± 0.081ab | 1.515 ± 0.158a | 8 | 0.444 ± 0.088abc | 0.380 ± 0.048a | 8 |
| Pedicel | ChS-2 | 15.843 ± 0.642bcde | 15.694 ± 1.205bcd | 13 | 1.465 ± 0.046b | 1.365 ± 0.052a | 13 | 0.597 ± 0.111ab | 0.740 ± 0.063a | 13 |
| F1 | ChS-2 | 21.543 ± 0.877abcd | 22.478 ± 0.773ab | 118 | 2.175 ± 0.054a | 2.258 ± 0.0484a | 60 | 0.685 ± 0.034a | 0.746 ± 0.027a | 60 |
| F2 | ChS-2 | 24.409 ± 1.573abc | 21.069 ± 1.203ab | 43 | 2.064 ± 0.128a | 2.208 ± 0.0481a | 25 | 0.486 ± 0.056abc | 0.465 ± 0.032a | 25 |
| F3 | ChS-2 | 24.947 ± 1.639ab | 24.364 ± 2.264ab | 21 | 2.260 ± 0.146a | 2.232 ± 0.128a | 12 | 0.420 ± 0.039abc | 0.500 ± 0.061a | 12 |
| F4 | ChS-2 | 18.149 ± 1.024abcd | 19.932 ± 0.904abc | 40 | 2.081 ± 0.117a | 2.095 ± 0.106a | 27 | 0.564 ± 0.074abc | 0.447 ± 0.035a | 27 |
| F5 | ChS-2 | 15.276 ± 1.460cde | 17.091 ± 1.569bcd | 20 | 1.683 ± 0.104ab | 1.653 ± 0.194a | 11 | 0.364 ± 0.057ab | 0.377 ± 0.076a | 11 |
| F6 | ChS-2 | 13.059 ± 0.859de | 15.525 ± 1.069bcd | 35 | 1.470 ± 0.079b | 3.218 ± 1.559a | 21 | 0.345 ± 0.047ab | 0.677 ± 0.374a | 21 |
| F7 | ChS-2 | 9.617 ± 0.813e | 11.374 ± 0.802cd | 14 | 1.220 ± 0.050b | 1.242 ± 0.047a | 12 | 0.269 ± 0.051b | 0.288 ± 0.036a | 12 |
| F8 | ChS-2 | 10.245 ± 0.864e | 10.235 ± 1.171d | 12 | 1.182 ± 0.034b | 1.192 ± 0.120a | 10 | 0.258 ± 0.039b | 0.281 ± 0.038a | 10 |
| average | ChS-2 | 19.247 ± 0.567 | 19.681 ± 0.509 | 316 | 1.891 ± 0.049 | 2.131 ± 0.190 | 191 | 0.519 ± 0.025 | 0.565 ± 0.048 | 191 |
| average | ChS | 19.478 ± 0.566 | 19.864 ± 0.510 | 324 | 1.884 ± 0.047 | 2.113 ± 0.185 | 199 | 0.515 ± 0.023 | 0.559 ± 0.047 | 199 |

**Notes.**
Values shown are the mean ± SE. Means in rows with same letters are not significantly different (we used Tukey HSD tests and assessed significance using a Bonferroni correction (alpha = 0.053 comparisons = 0.017)).

## Antennal rhinaria (AR)

Antennal rhinaria (AR) were further divided into two subtypes based on their morphology. Subtype 1 (AR-1) was distinguished by a thin walled multi-porous surface covering a pit from which a multi-porous protrusion was found (Fig. 5E). Subtype 2 (AR-2) was similar to the cavity sensilla, with a large opening and sensory peg inside (Fig. 5F). AR were distributed across the ventral view of the psyllid antenna. The AR located on F2 were subtype 1 in both sexes, and we found both subtypes on F4 and F6. Two sensilla basiconica (SB-1 and SB-2) protruded next to or from the AR on F7 (Fig. 5G). AR-1 was 19.250 ± 1.488 μm long for females and 15.929 ± 1.481 μm long for males, and 13.896 ± 1.623 μm wide for females and 11.331 ± 0.826 μm wide for males. The inside diameter of the opening of AR-2 was 7.049 μm for females and 10.368 μm for males. The protrusion at the bottom of the AR was a SW sensilla with a porous surface (Fig. 3F). Using TEM analysis, we found that the SW protrusion in AR increased in size and DOS became more numerous going from the upper part to the lower part (Figs. 3–5F). The porous surface and DOS found in AR was indicative of chemoreceptor function.

## Cavity sensilla (CvS)

Cavity sensilla (CvS) were pit organs with thin walls and pegs. CvS could only be found on the anterior view of the apical part of F8. We identified two subtypes of CvS (CvS-1 and CvS-2) on *B. gobica* antenna based on their location and morphology. CvS-1 and CvS-2 were located at the base of LAS and SAS, respectively, on flagellomere F8 (Figs. 5C–5D).

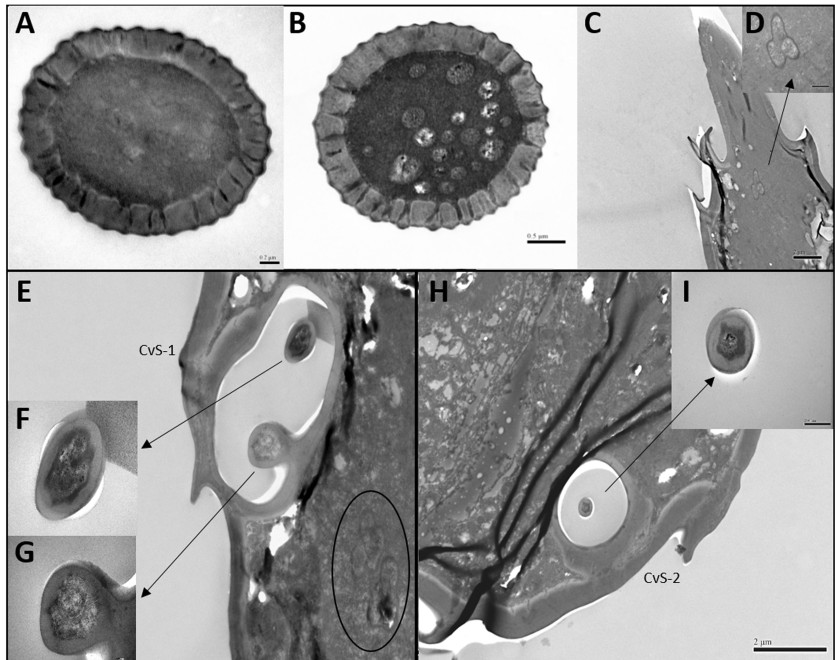

**Figure 6 Transmission electron micrographs of various antennal sensilla of *Bactericera gobica*.** (A–C) Cross section of the distal (A) and basal (B) part of the short apical setae (SAS) and a tilted transect through the SAS and the tip of the flagellomere. Twenty-one dendritic outer segments (DOS) were found in female SAS (B). The black arrow in the graph points to a group of dendrites innervating SAS and are enveloped by dendritic sheath. (E–G) Cross section through the cavity sensilla subtype 1 (CvS-1) below the long apical setae (LAS). Two pegs were found at the bottom of CvS-1 on female *B. gobica* antenna. The upper peg is aporous (F), and the lower peg is porous (G). The black circle in the graph shows two groups of dendrites innervating LAS and are enveloped by dendritic sheath. (H–I) Cross section through the cavity sensilla subtype 2 (CvS-2) below the SAS. There is only one aporous peg at the bottom of CvS-2 (I).

The widest diameter of the opening of CvS-1 was about 1.757 μm for females and 1.519 μm for males. The opening of CvS-2 was slightly larger than CvS-1, with an average inside diameter of 2.198 μm for females and 1.729 μm for males (Table 4). Using TEM analysis, we found two pegs at the bottom of CvS-1 in females (Figs. 6E–6G). In contrast, when using SEM analysis, only one peg-shaped sensilla was observed (Fig. 5C). These two pegs were both single walled, and only one wall had pores (Figs. 6E–6F). In contrast, male CvS-1 and CvS-2 in both sexes had only one aporous peg at the bottom (Figs. 6H–6I).

## Sensilla campaniform (SCA)

Sensilla campaniform (SCA) were oval shaped and only found on the dorsal view of pedicel segments in both sexes (Fig. 5B). Using SEM analysis, neither pores nor openings were found in SCA. The SCA averaged 7.643–8.154 μm long and 6.558–6.411 μm wide and had a smooth surface. We were unable to examine the cross sections of SCA with TEM because the SCA is situated dorsally on the pedicel segments.

### Differences between female and male antennal sensilla

The external morphology of sensilla on the antennae of female and male *B. gobica* were similar (Table 2). We found no significant differences between the lengths of any segments (Table 1). We also observed no differences in sensilla types between females and males (Table 4). However, several differences in the quantity, array, and inner structure of sensilla were found between the sexes. The basal width of LAS on male antennae was significantly larger than that of the female antennae (Two-sample $t$-test, $t_{20} =13.617$, $p =0.012$). ChS on flagellomeres F1, F3, F6, and F7 in males were greater in number than those in females (Table 2 and Figs. 4A–4H). ChS were also more numerous in males than in females. Moreover, DOS found in the protrusion of AR were more abundant in males than in females (Figs. 4M and 4N). In contrast, the number of ChS on two of the antennal segments (pedicel and flagellomere F4) and the DOS in the apical setae were more abundant in females than males. Finally, an additional porous peg was identified in the CvS-1 of females (Figs. 4O and 4P).

## DISCUSSION

The segmentation and morphology of *B. gobica* antennae were quite similar to other psyllid species described in prior studies, including carrot psyllid *T. apicalis*, citrus psyllid *Trioza erytreae*, pear psyllid *C. bidens*, and potato psylla (*Arras, Hunter & Bextine, 2012*; *Kristoffersen et al., 2006*). In total, we found seven types and twelve subtypes of sensilla on *B. gobica* antennae. The distribution of antennal sensilla in both sexes of *B. gobica* was nearly asymmetrical. The color of the flagellomere 6–8 of *B. gobica* antennae was darker than the remaining segments, especially at the start of the antennal rhinaria. *Arras, Hunter & Bextine (2012)* reported similar findings for the Asian citrus psyllid *D. citri*. Consistent with previously studied psyllids, goji berry psyllid *B. gobica* antenna had 10 segments, including one scape segment, one pedicel segment, and eight flagellomeres, ranging in total length from 940–1004 µm. *T. erytreae* has a similar antennae length to *B. gobica* (about one mm long) (*Moran & Buchan, 1975*). However, some psyllid species have shorter antenna than *B. gobica,* such as *D. citri* (440–450 µm) and the carrot psyllid *T. apicalis* (595 µm) (*Arras, Hunter & Bextine, 2012*). *B. gobica* also has longer apical setae (48.84–51.31 µm) than *T. apicalis* (25.81 µm) and *D. citri* (18.87 µm).

Psyllid antenna have chemoreceptive function. The combination of behavior, anatomy, genomics, and electrophysiology techniques suggest that psyllids use chemical cues, most likely sensory arrays on the antennae, to locate host plants and mates (*Yuvaraj et al., 2013*; *Qi, Li & Guo, 2021*). Thus, we expected to find some chemosensory sensilla on the antennae. Antennal sensilla with olfactory function usually have porous surfaces, sensillar channels, and inner dendritic segments that enable the reception of various odorous stimuli by the sensilla (*Fernandes et al., 2020*; *Jeong et al., 2020*; *Yan et al., 2017*; *Gao, Luo & Hammond, 2007*). The dendritic outer segments (DOS; the portion above the ciliar region of the sensory cells) are important indicators of the function of the sensilla (*Hallberg, Hansson & Lfstedt, 2003*). Using TEM and SEM, we showed that five sensillar types (apical sentae, cavity sensilla, antennal rhinaria, sensilla basiconica and chaetica) on *B. gobica* antenna likely have olfactory function.

The two apical setae, long apical setae (LAS) and short apical setae (SAS) have had various names in prior studies including "distal antennal sensory pegs" (*Arras, Hunter & Bextine, 2012*), "terminal hairs", or "bristles", (*Zheng et al., 2020*; *Kristoffersen et al., 2006*), or other terms. This has made comparison of function difficult, but LAS and SAS have generally been suggested to have olfactory functions due to their porous surfaces (*Zheng et al., 2020*; *Kristoffersen et al., 2006*). Nine DOS were first reported in the apical setae of the carrot psyllid *T. apicalis* (*Kristoffersen et al., 2006*). We found more DOS in *B. gobica* than was reported for the carrot psyllid *T. apicalis*: 18 and 45 DOS in the lumen of female and male LAS, respectively. *Kristoffersen et al., (2006)* reported the LAS of carrot psyllid *T. apicalis* was innervated by three groups of receptor cells. In contrast, we found LAS and SAS were innervated by two groups and one group of receptor cells, respectively. These findings are consistent with previous studies on fruit flies that receptor neuron cells innervated olfactory antennal sensilla that are usually divided into many brush-shaped dendritic branches in the ciliar region (*Liu et al., 2021*; *Kaupp, 2010*). Altogether, our results suggested that the apical setae are chemoreceptive sensilla in *B. gobica*, considering both internal and external structures of LAS and SAS.

Cavity sensilla (sometimes called sensilla coeloconica) have been found on the antenna of both adults and larvae in many psyllid species. Many different types of pegs have been found in insect cavity sensilla (*Schneider, 1964*). We found two cavity sensilla (CvS-1 and CvS-2) at the base of LAS and SAS on *B. gobica*. Cavity sensilla are believed to aid in perception of temperature, humidity, and $CO_2$, despite the pegs being hidden beneath the antennal surface (*Zheng et al., 2020*; *Kristoffersen et al., 2006*).

Antennal rhinaria —also called placodea sensilla, partitioned sensory organs (PSOs), and cavity sensillum in previous studies of psyllids —are suggested to be equivalent to plate organs in aphids (*Kristoffersen et al., 2006*). Antennal rhinaria are thought to be the principal odorant sensors of several psyllid species that sense plant volatiles (*Kristoffersen, Larsson & Anderbrant, 2008*; *Yuvaraj et al., 2013*; *Coutinho-Abueu et al., 2014*). Flagellomeres 2, 4, 6, and 7 of goji berry psyllids each contain one antennal rhinaria. Antennal rhinaria of *B. gobica* could be further divided into two subtypes based on morphology. *Arras, Hunter & Bextine, 2012* reported similar findings for potato psyllid *B. cockerelli*. Interestingly, no obvious DOS were found at the tip of antennal rhinaria nor the apical setae. However, DOS were found in cross-sections of the lower parts of the apical setae and antennal rhinaria (Figs. 6A–6B). Similar findings that DOS occur within the middle portion of the digitiform organ have been reported on sensory organs of the scarab beetle *Melolontha melolontha* and fruit fly *Bactrocera dorsalis* (*Eilers et al., 2012*; *Liu et al., 2021*). The multiparous surface and numerous DOS in the cross-sections of *B. gobica* antennal rhinaria clearly suggest the olfactory function of this sensilla type in *B. gobica*, which is consistent with previous studies on other psyllid species (*Zheng et al., 2020*; *Onagbola et al., 2008*; *Kristoffersen et al., 2006*). The olfactory function of this sensilla type has been clearly demonstrated using single-unit electrophysiology. Further, it is the only psyllid antenna sensilla type that has been verified to have olfactory function so far. Studies conducted on carrot psyllid *T. apicalis* have shown that antennal rhinaria are innervated by three sensory cells with branched DOS (*Kristoffersen, Larsson & Anderbrant, 2008*). The three olfactory receptor neurons (ORNs)

detected in each of Asian citrus psyllid *D. citri* AR reveal repeated responses to citrus-related odors. These neurons in *D. citri* even have the potential to detect and discriminate a large variety of odors and blends (*Coutinho-Abueu et al., 2014*).

We only found sensilla basiconica on the F7 segment of *B. gobica*. SBs have been called "haired shaped sensillum" in carrot psyllid *T. apicalis* studies, and they were also found on the antenna of third-, fourth-, and fifth instar nymphs of *D. citri* (*Zheng et al., 2020*; *Kristoffersen, Larsson & Anderbrant, 2008*). The two SBs (SB-1 and SB-2) on *B. gobica* had grooved surfaces, each with a blunt tip that resembled apical setae. *Kristoffersen, Larsson & Anderbrant (2008)* reported that SB sensilla were innervated by three sensory cells in carrot psyllid *T. apicalis*. However, we found more DOS in SB-1 and SB-2 of *B. gobica* than was reported for carrot psyllid (*Kristoffersen et al., 2006*). Schneider1964 reported that sensilla basiconica are the most common and important chemoreceptors on the insect antennae. Our results likewise suggested that sensilla basiconica have chemoreceptory function in *B. gobica*.

Sensilla chaetica is another sensilla type that we suggest has olfactory functions on *B. gobica* sensilla. ChS is the most widely distributed type on the antenna of *B. gobica*. We found cuticular channels and a DOS in the septilateral and octagon cross-sections of *B. gobica* ChS. Our findings differed from the electron dense material of corresponding sensilla on carrot psyllid *T. apicalis* and citrus psyllid antenna (*Kristoffersen et al., 2006*; *Onagbola et al., 2008*). Similar findings that sensilla chaetica may function as olfactory sensors based on TEM observation have been reported for the pine weevil *Pissodes nitidus* (*Yan et al., 2011*). *Schneider (1964)* pointed out that the nerve fibers mostly end in the tip of sensilla chaetica, showing their promise of having olfactory function.

In addition, we found two mechanosensory sensilla in *B. gobica* on the scape and pedicel segments: sensilla trichodea (ST) and sensilla campaniform (SCA). Even though ST is usually considered to have olfactory function (*Schneider, 1964*), no sensillar channel was observed in the cross section of *B. gobica* ST. This suggests that the *B. gobica* ST is unlikely to be an olfactory sensilla. *Onagbola et al. (2008)* also reported many aporous ST on the scape and pedicel segments of *D. citri* adults. The location and morphology of sensilla campaniform in *B. gobica* is consistent with other psyllids and aphids (*Soroker et al., 2004*). However, we did not find the intracuticular sensillum reported in carrot psyllid *T. apicalis* on *B. gobica* antenna.

It is worth noting that we found several differences in the morphology and internal structure between sensilla chaetica, cavity sensilla, apical setae, and antennal rhinaria of the two sexes. Sex pheromones have been identified for the Asian citrus psyllid *D. citri* (*Zanardi et al., 2018*), pear psyllid (both *C. pyricola* and *C. bidens*), and potato psyllid *B. cockerelli* (*Guedot, Hiorton & Landolt, 2010*). There are also some variations in the response of male and female blue gum psyllid *Ctenarytaina eucalypti* OSNs to different plant volatiles (*Yuvaraj et al., 2013*). *Nicolas et al. (2020)* reported that sexual dimorphism, particularly at the level of sensory organs, is usually attributable to sexual selection. This is reflected in antennae being notably developed in males of species that need to detect a sex pheromone. For instance, male aphid midges *Aphidoletes aphidimyza* have longer and more highly developed antennae than females, and females emit a sex pheromone for mating. Similarly,

we found that *B. gobica* males tended to have longer antenna, though the difference was not statistically significant. Male *B. gobica* had more ChS and DOS in their AR than females in our study. These findings might suggest the presence of a sex pheromone in *B. gobica*.

In general, the sensilla set-up of *B. gobica* is similar to other psyllids. Because we found that *B. gobica* has longer antenna with more DOS compared to other psyllids (*Kristoffersen et al., 2006*; *Arras, Hunter & Bextine, 2012*), *B. gobica* might have at least the same or even stronger olfactory function and sensitivity compared to other psyllids. Homoptera have comparatively simple olfactory systems and lack antennal lobe structures, even though they apparently depend on long-range olfactory orientation (*Coutinho-Abueu et al., 2014*). The olfactory systems of the Psylloidea seem particularly small even for Homoptera (*Chapman, 1982*; *Kristoffersen et al., 2006*). The sparse sensilla on psyllid antenna may require rather high concentrations of odor stimuli to respond. However, prior study demonstrated that the small olfactory system of the citrus psyllid is efficient at covering a vast odor space using as few as 10 ORNs (*Coutinho-Abueu et al., 2014*). It also showed a high degree of neuronal redundancy in the carrot psyllid *T. apicalis* (*Coutinho-Abueu et al., 2014*). A small and specialized olfactory setup may be sufficient for psyllids for several reasons. First, psyllids are very small organisms, and the viscous properties of the air are pronounced around small structures (*Kristoffersen et al., 2006*). Second, organisms are likely to evolve unique olfactory systems that contribute to detecting behaviorally relevant volatiles from their habitats (*Coutinho-Abueu et al., 2014*). Like some other psyllids, the goji berry psyllid *B. gobica* is a host specialist so may not need an elaborate olfactory system. The reduced olfactory system we observed in our study could be related to *B. gobica'* s host plant specialization.

SEM and TEM are both valuable tools in biological and physical research. The main difference between SEM and TEM is that SEM creates images using electrons to scan samples' surfaces, while TEM creates images using electrons to pass through samples. As a result, studies on insect antennal sensilla that use SEM alone only provide information on the external structure of sensilla. To date, there are very few studies of psyllid antennae, particularly those that include TEM data needed to examine internal structures. Prior work has generally found either no sexual dimorphism or only a single difference in the psyllid sensillar setup between sexes (*Onagbola et al., 2008*; *Kristoffersen et al., 2006*; *Arras, Hunter & Bextine, 2012*). However, using both SEM and TEM, we observed several differences in the sensilla setup and internal structures between sexes. Moreover, we found more DOS in apical setae than rhinaria sensilla, suggesting an important new direction for future work. We suggest future studies use single-unit electrophysiology and gas-chromatograph-linked SSR to examine the sensilla we found with promising olfactory function (such as apical setae). Indeed, studies using these approaches can verify the olfactory function of the sensilla described in our study.

## CONCLUSIONS

Altogether, we have comprehensively revealed the fine morphology of the antennae of *B. gobica*, highlighting differences and similarities between sexes. We also compared the

typology and the distribution of antennal sensilla of *B. gobica* with prior work on other psyllid species. To understand the olfactory specificity and sensitivity of the goji berry psyllid, further behavioral and electrophysiological studies will be needed. Findings in this study complement the knowledge gap in the olfactory perception of goji berry psyllid and have the potential to be used in the analysis of the function of the various sensilla on psyllid antennae. Understanding the morphology of psyllid antennae is the first step towards understanding olfactory specificity and sensitivity needed to develop and implement effective, sustainable pest control strategies that leverage olfactory disruption.

## ACKNOWLEDGEMENTS

The authors would like to thank the Electron Microscopy Laboratory at the Institute of Food Science and Technology (CAAS) for TEM technical support and the Medical Experimental Center of the China Academy of Chinese Medical Sciences for SEM technical support.

### Funding

This research was funded by the Fundamental Research Funds for the Central Public Welfare Research Institutes (project: ZZ14-YQ-048 & ZZXT202009), the Development of application technology of attractant and control agent for agricultural piercing and sucking pests in Ningxia (Project: NGSB-2021-10-02), and the Scientific and technological innovation project of China Academy of Chinese Medical Sciences(Project: CI2021A03906). The funders had no role in study design, data collection and analysis, decision to publish, or preparation of the manuscript.

### Grant Disclosures

The following grant information was disclosed by the authors:
The Fundamental Research Funds for the Central Public Welfare Research Institutes (project: ZZ14-YQ-048).
The Fundamental Research Funds for the Central Public Welfare Research Institutes (project: ZZXT202009).
Development of Application Technology of Attractant and Control Agent for Agricultural Piercing and Sucking Pests in Ningxia (Project: NGSB-2021-10-02).
The Scientific and Technological Innovation Project of China Academy of Chinese Medical Sciences (Project: CI2021A03906 ).

### Competing Interests

The authors declare there are no competing interests.

### Author Contributions

- Yang Ge conceived and designed the experiments, prepared figures and/or tables, authored or reviewed drafts of the paper, and approved the final draft.

- Olivia M. Smith and Pingping Liu conceived and designed the experiments, authored or reviewed drafts of the paper, and approved the final draft.
- Weilin Chen performed the experiments, prepared figures and/or tables, and approved the final draft.
- Qingjun Yuan, Chuanzhi Kang, Tielin Wang, Jiahui Sun and Binbin Yan analyzed the data, prepared figures and/or tables, and approved the final draft.
- Xiaoli Liu and Lanping Guo conceived and designed the experiments, prepared figures and/or tables, and approved the final draft.

## Data Availability

The raw data is available in the Supplementary File.

## Supplemental Information

Supplemental information for this article can be found online at http://dx.doi.org/10.7717/peerj.12888#supplemental-information.

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
