# Peer review of "Morphological characterization and sexual dimorphism of the antennal sensilla in Bactericera gobica Loginova (Hemiptera: Psyllidae)—a scanning and transmission electron microscopic study"

_PeerJ, doi:10.7717/peerj.12888_

## Round 0.1 · original submission · Minor Revisions

Dear authors,

Your article has been reviewed by 3 peer reviewers. The reviewers have provided evaluations and made recommendations for revisions to your manuscript. The reviewers have raised some concerns about your methodological approaches and interpretations, which need to be carefully considered. Reviewers 1 and 3 have concerns that if ultrasonic cleaning can damage the structure of antenna. Reviewer 2 has a question about data presentation in the tables. Reviewer 3 has commented that the methodology lacks sufficient details and also attached a review report file that should be addressed.

I invite you to respond to the reviewers' detailed comments and revise your manuscript. All three reviewers' comments need to be addressed before the manuscript can be accepted.

Thank you for submitting your manuscript to PeerJ and I look forward to receiving your revision.

Best,

Xiaotian Tang, Ph.D.
Academic Editor, PeerJ
xiaotian.tang@yale.edu

Reviewer 1 ·

Basic reporting

In this manuscript authors compared the morphological characterization of the antennal sensilla in Bactericera gobica Loginova using SEM and TEM. This study provides some useful information for the future studies on defining the olfactory function of psyllid antenna.

Experimental design

The experimental design is well supported the result and their analysis.

Validity of the findings

The authors found some differences between female and male antennal sensilla.

Additional comments

1. The reason why the authors chose TEM?The difference between TEM and SEM should be added in discussion, to highlight the differences and characteristics between this paper and other studies.
2. Why did the authors only chose 3-day-old adults for the experiment?
3. In line 111, "the entire bodies of the B. gobica specimens were cleaned two times each in 70% ethanol using ultrasonic waves". Therefore, whether ultrasonic cleaning can damage the structure of antenna and affect the experimental results.
4. The whole experiment is only a description of morphology. If there is relevant functional verification in the follow-up experiment, the research may be more meaningful.
5. Manuscript format needs to be checked. For example, In Line116 and 367...

Reviewer 2 ·

Basic reporting

This is a well-written manuscript.

Experimental design

The research question is well defined. This study is the first to report the morphology of B. gobica antennal sensilla with the aid of SEM and TEM.

Validity of the findings

The statistical analysis used is appropriate. The conclusions are well stated.
However, there is one issue to be addressed. Why the size (mean ± SE) of sensilla chaetica (ChS) presented in the Tables 4 and 5 were different. Please clarify it?

Additional comments

The title reflects the contents of the manuscript. All figures are easily readable with sufficient resolution.

Reviewer 3 ·

Basic reporting

no comment

Experimental design

no comment

Validity of the findings

no comment

Annotated reviews are not available for download in order to protect the identity of reviewers who chose to remain anonymous.

---

## Round 0.2 · accepted · Accept

Dear Authors,

Thank you for adequately addressing all the concerns raised during the initial review. I am pleased to inform you that your article, "Morphological characterization and sexual dimorphism of the antennal sensilla in Bactericera gobica Loginova (Hemiptera: Psyllidae)—A scanning and transmission electron microscopic study ", has now been accepted for publication in PeerJ.

Thank you for your submission and we hope you will continue to support PeerJ.

Best,

Xiaotian Tang

Reviewer 3 ·

Basic reporting

I went through the revised version of your manuscript, which has been greatly improved. The revised article meets the PeerJ criteria and can be accepted as is.

Experimental design

no comment

Validity of the findings

no comment

Additional comments

no comment